

# Predicting social and health vulnerability to floods in Bangladesh

Donghoon Lee[1,2], Hassan Ahmadul[3], Jonathan Patz[4], and Paul Block[1]

[1]Department of Civil and Environmental Engineering, University of Wisconsin-Madison, Wisconsin, USA
[2]Climate Hazards Center, Department of Geography, University of California, Santa Barbara, California, USA
[3]Red Cross Red Crescent Climate Centre, The Hague, the Netherlands
[4]Global Health Institute, Nelson Institute Center for Sustainability and the Global Environment, and Department of Population Health Sciences, University of Wisconsin-Madison, Wisconsin, USA

**Correspondence:** Donghoon Lee (dlee@geog.ucsb.edu)

**Abstract.** Floods are the most common and damaging natural disaster in Bangladesh, and the effects of floods on public health have increased significantly in recent decades, particularly among lower socio-economic populations. Assessments of social vulnerability on flood-induced health outcomes typically focus on local to regional scales; a notable gap remains in comprehensive, large-scale assessments that may foster disaster management practices. In this study, socio-economic, health, and coping

capacity vulnerability and composite social-health vulnerability are assessed using both equal-weight and principal component approaches using 26 indicators across Bangladesh. Results indicate that vulnerable zones exist in the northwest riverine areas, northeast floodplains, and southwest region, potentially affecting 42 million people (26% of total population). Subsequently, the vulnerability measures are linked to flood forecast and satellite inundation information to evaluate their potential for predicting actual flood impact indices (distress, damage, disruption, and health) based on the immense August 2017 flood event.

Overall, the forecast-based equally weighted vulnerability measures perform best. Specifically, socio-economic and coping capacity vulnerability measures strongly align with the distress, disruption, and health impacts records observed. Additionally, the forecast-based composite social-health vulnerability index also correlates well with the impact indices, illustrating its utility in identifying predominantly vulnerable regions. These findings suggest the benefits and practicality of this approach to assess both thematic and comprehensive spatial vulnerabilities, with potential to support targeted and coordinated public

disaster management and health practices.

## 1 Introduction

Public health outcomes stemming from flood events are typically acute and severe, particularly in developing or tropical regions, potentially including death and injury, contaminated drinking water, endemic and infectious diseases, and community disruption and displacement. Although the impacts of floods on public health have been investigated (Ahern et al., 2005;

Alderman et al., 2012; Batterman et al., 2009; Du et al., 2010; Tapsell et al., 2002), integrated management of flood and health risks is technically and institutionally limited.

Unsurprisingly, public health research on the impacts of natural disasters predominantly focuses on clinical, microbiological, and ecological aspects, including vaccines, therapy, and improved treatment (Colston et al., 2020; Schwartz et al., 2006).





Development of flood prediction and disaster management has principally targeted advancing climate and hydrologic aspects,
with much less focus on considerations of health vulnerability, community risk, and early warning systems, particularly in
developing countries (Kovats et al., 2003). This often results in under-prediction of event outcomes on marginalized and
susceptible communities (WHO, 2013). Only recently has the global community started calling for multi-sectoral disaster
forecast and warning systems to support integrated disaster management (UNISDR, 2015), including novel indicators of public
health risk and vulnerability. In addition to prioritizing forecast-informed health risks, identifying vulnerable regions and
populations to establish targeted and coordinated public health practices is critical (Akanda et al., 2011).

In Bangladesh, floods are the most significant and damaging natural disaster. A vast majority (75%) of the country is within
10 meters above mean sea level, and an even higher fraction of landmass (80%) is classified as floodplain (BBS, 2018).
Approximately 78% of Bangladesh's total population, mostly rural and poor, live in floodplain regions BBS (2016). On average,
18% of the country is inundated in any given year. Catastrophic events have occurred most recently in 1988, 1998, 2007 and
2017, affecting 60% of the nation for nearly 3 months and causing erosion, landlessness, mortality, environmental refugees,
destruction of property and crop lands, and disruption of communication and health systems BBS (2018); CEGIS (2003). In
addition, these floods have led to outbreaks of water-borne disease and epidemics as a result of contaminated drinking water.
The effects of floods on diarrheal diseases have been a major public health concern, as diarrheal disease is one of the leading
causes of morbidity and mortality, particularly among people with low socio-economic status and poor sanitation (Alderman
et al., 2012; Kosek et al., 2003); risks have significantly increased in recent decades (Hashizume et al., 2008). These conditions,
when combined with social inequity, low literacy rates, deprivation, and insufficient institutional capacity, can lead to precarious
situations (Mazumder et al., 2015; Shahid, 2010; Mani and Limin Wang, 2014).

In this regard, a number of studies investigate health vulnerability to floods based on demographic and socio-economic
factors in Bangladesh. Kunii et al. (2002), for example, associate the impacts of the 1998 Bangladesh flood on community
health, such as fever, diarrhea, and respiratory problems, with socioeconomic status, water handling, and household sanitation.
Hashizume et al. (2008) examine flood-related diarrheal incidents during the 1998 flood in Dhaka and discover substantially
higher flood-related cases in the post-flood period for populations with low socio-economic status and weak sanitation and
hygiene facilities. Schwartz et al. (2006) analyze demographic and clinical data of patients with flood-related diarrhea in Dhaka
and find an increase in flood-related epidemics in populations with low socioeconomic status, inferior sanitation, dwellings in
flood-prone areas, and minimal access to care. Shahid (2010) discusses direct and indirect impacts of climate change on public
health and highlights high poverty rates and limited access to sanitation facilities in Bangladesh as significantly impacting
diarrheal-related health problems. Nahar et al. (2014) note that women and poor populations in Bangladesh are especially
vulnerable to poor mental health status in the post-flood period.

While these studies illustrate the underlying relationships between demographic and socio-economic vulnerability factors
and flood-induced health risks at specific locations (e.g., Dhaka), there is no explicit link to disaster management practices
for more rural and poor regions. Presumably, if such vulnerability factors are aggregated and concurrently evaluated with
physical flood information to estimate at-risk populations, this collective information may be strategic for informing pre- and





post-flood disaster management plans. This motivates evaluation of combined socio-economic and health factors to develop comprehensive and practical vulnerability metrics.

Thus, we develop a social-health vulnerability (SHV) indicator based on a variety of demographic, socio-economic, health, and infrastructural indicators for all of Bangladesh to identify the most vulnerable regions, means for vulnerability/risk reduction, and to enhance response capacity and efficiency for international and local disaster management agencies. We also examine the predictability of flood impacts on livelihood, community, and health sectors by linking vulnerabilities to flood forecasts and satellite inundation for the catastrophic 2017 Bangladesh flood event. While the emphasis here is on the impacts

of flood on social and health, multi-sectoral risk warning systems, coupled with vulnerability and risk characteristics, can be envisioned.

## 2   Data

Here we describe the data necessary to establish vulnerability indicators, including survey and census data, spatially explicit flood forecasts, satellite inundation, and population data. Flood impact records for the 2017 August event from post-disaster

reports are also presented.

### 2.1   Survey, census, and population data

A large number of relevant data, resources, and documents are available across various governmental agencies, including the Bangladesh Bureau of Statistics (BBS), Department of Disaster Management (DDM), and Directorate General of Health Services (DGHS). These agencies typically report outcomes for administrative units defined as (large-to-small): division, district,

Upazila, and union.

    A census is conducted in Bangladesh approximately every 10 years, with 2011 being the most recent. Upazila-scale population, household census data, and various demographic and socio-economic records are available via the BBS (BBS, 2015). This census data provides a significant proportion of the indicators of socio-economic vulnerability used in this study. Poverty estimates (population below the upper poverty line) in 2010 measured by the World Bank and BBS in conjunction with World

Food Programme (WFP) are also obtained from the BBS.

    Information outlining comprehensive measures of coping capacity at local scales is often incomplete or not available. In Bangladesh, the BBS conducts household surveys and quantifies disaster-related statistics for twelve main natural disasters (BBS, 2016). From this report, we adopted district-level statistics to represent coping capacity and public health vulnerability indicators for flood disasters. Examples include knowledge and perceptions of disasters (population assume that natural process

causes critical disasters), damages and losses, households receiving financial support, population lacking safe drinking water, etc.

    Additionally, health facility (e.g., location, capacity, etc.) and physician data are obtained from the Facility Registry (http://facilityregistry.dghs.gov.bd) and the Health Dashboard (https://dghs.gov.bd/index.php/en/home), respectively. From this data, the number of hospital beds and physicians are estimated to reflect the capacity of health system and health workforce of





each Upazila. The national average of hospital beds per 1,000 people and physicians per 10,000 people in 2019 are measured
as 0.6 (0.8 in 2015 by World Bank) and 0.58 (0.53 in 2017 by World Bank), respectively.

For spatial population data, the WorldPop population per pixel data in 100m resolution is obtained and rescaled linearly with
a World Bank population record of 2017 (World Bank, 2018; Worldpop et al., 2018).

## 2.2    Flood forecast, satellite inundation, and population data

In Bangladesh, the Flood Forecasting and Warning Centre (FFWC) provides flood forecasts and warning services country-
wide. FFWC's flood forecasting system is based on the MIKE 11 model, a one-dimensional water modeling software for the
simulation of water levels and discharges in river networks and flood plains. Two-dimensional flood inundation (flood depth)
forecasts are created using Digital Elevation Models (DEM) at 300 m spatial resolution. The current early flood warning system
offers a 120 hour lead-time (FFWC, 2018). The FFWC acknowledges that flood forecasts may underestimate or overestimate

inundation depths and extent given the lack of model updates and course spatial resolution. These FFWC issued flood forecasts
are utilized for the August 2017 event (issued August 16th) evaluated here. These forecasts were verified by FFWC with
observed inundation maps from Sentinel-1 satellite images, illustrating good agreement in the northwestern and northeastern
regions (FFWC, 2018). We obtained the satellite inundation data for the August 2017 flood event generated using Sentinel-1
Synthetic Aperture Radar images (August 22nd, 24th, 27th, and 29th) from the International Centre for Integrated Mountain

Development (Uddin et al., 2019) (Figure S1).

## 2.3    Flood impact records

The Global Shelter Cluster has aggregated relevant post-disaster reports and data for the August 2017 flood event in Bangladesh
through government agencies and international relief organizations (https://www.sheltercluster.org/response/bangladesh-mon
soon-floods-2017). Specifically, we leverage the 72-hour Rapid Assessment report published August 21st, the flood damage

data reported on September 3rd by the DDM and Natural Disaster Response Coordination Group, and monthly hazard incident
report from the Network for Information, Response and Preparedness Activities on Disaster (NIRAPAD) (NIRAPAD, 2017b).
The DGHS reported health outcomes from the August 2017 flood collected between July to September. From this, we extract
the number of diarrheal incidents and other adverse health outcomes, including incidents of respiratory tract infections (RTI),
eye and skin diseases, snake bites, drowning, and other injuries.

## 115    3    Methods

Spatially explicit vulnerability and risk maps can support decision-makers by enhancing their ability to take appropriate actions.
However, vulnerability assessment is complicated by environmental, social, economic, and political patterns of societies. To
date, no standard model or methodology exists to guide spatial vulnerability assessments for natural disasters, although the
number of related studies is rapidly increasing (Villagrán de Léon, 2008; Ward et al., 2020). In this study, we select socio-

economic, health, and coping capacity vulnerability domains consisting of 26 indicators based on the literature, availability of





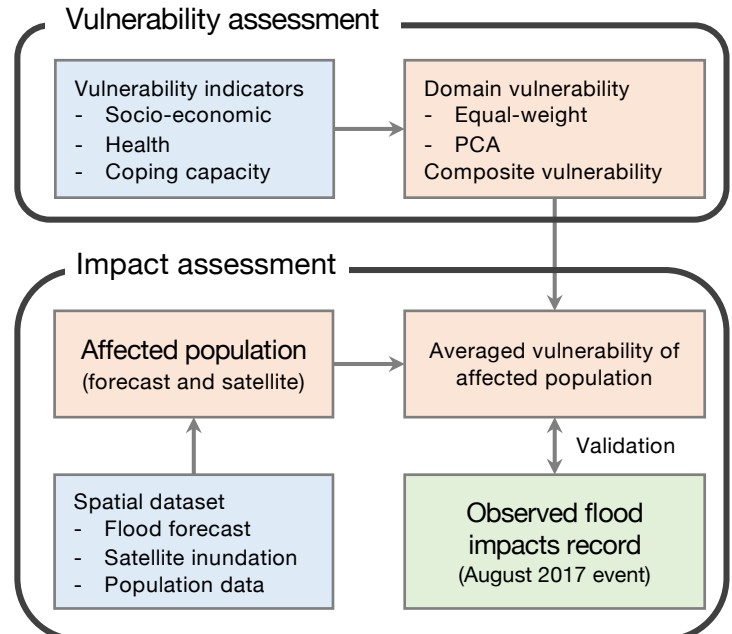

**Figure 1.** Vulnerability and impact assessment framework.

data, and their vulnerability influences. The domain-level vulnerability (DV) is estimated using two approaches of vulnerability calculation, namely equal-weights and Principal Component Analysis (PCA); social-health vulnerability (SHV) is measured using equal weights. The flood forecast and satellite inundation information are applied to estimate affected population during the August 2017 flood event. Finally, vulnerability of the affected population is assessed and validated against records of

post-disaster flood impacts. The overall framework of this study is illustrated in Figure 1.

### 3.1    Social-health vulnerability assessment

### 3.1.1    Domain and Indicators selection

Previously, assessments of spatial vulnerability conditioned on socio-economic factors have been conducted for a number of regions of Bangladesh (Ahsan and Warner, 2014; Dewan, 2013; Gain et al., 2015; Hoque et al., 2019; Rabby et al., 2019; Roy

and Blaschke, 2015) and more broadly for the entire country (DDM, 2017; Islam et al., 2013). Method of assessment, indicators, study area, scale, and data are summarized in Table 1. These studies typically select vulnerability domains and indicators based on the context of the target disaster and study area or from a pre-defined approach in the literature. In previous studies, the domains include socio-economic, adaptive or coping capacity, and unique exposure or hazard domains, such as agricultural, physical (climate, flood, or coastal hazard), and infrastructure. For vulnerability models, an addictive model (equal weights) or

analytic hierarchy process analysis (AHP) (custom weights from stakeholder engagement or expert opinion (Saaty and Vargas,





2012)) are most common. A PCA analysis (e.g. Cutter et al. (2003)), is also frequently employed to identify dominant spatial patterns and to generate a composite vulnerability. The majority of studies adopt the equal weights approach such that each domain contributes equally to the composite vulnerability.

In this study, the SHV includes 26 indicators along with three indicator domains: socio-economic (15 indicators), health (5 indicators), and coping capacity (6 indicators) domains. The SHV specifically precludes physical indicators (e.g., low elevation, proximity to river, etc.), as flood hazard information (i.e., flood inundation) will be linked later through the impact assessment. Instead, we include a health domain uniquely reflecting flood-induced health risk that have rarely been considered in previous studies. Indicators are selected on the basis of their relevance to each domain vulnerability and availability of data for the country at Upazila or district level (Table 2).

The socio-economic domain broadly represents the potential impact of the hazard conditioned on the existing societal context. Based on the literature review, we select 15 indicators relevant to demographic (3), built environment (5), social (4), and economic (3) categories, drawing on the most recent population census data. Comparatively, the coping capacity domain represents the ability to cope with or adapt to the hazard. In the literature, coping capacity indicators are surveyed for the local region, or proxy data from the census are used, such as households with communication devices and vehicles, literacy rates, education levels, etc. For this study, we apply 6 indicators specifically measured to represent the level of disaster resilience in each district across Bangladesh, including: 1) percentage of households affected by floods, 2) percentage of children did not attend to school due to disasters, 3) percentage of household have not taken disaster preparedness activities, 4) percentage of population with knowledge and perception about disaster, 5) percentages of households received financial support from agencies, and 6) ratios of total damage/loss to total income (Table 2). In Bangladesh, several studies and reports investigate appropriate health indicators in the context of disaster management (DGHS, 2018; Schwartz et al., 2006; Shahid, 2010; WHO, 2013). However, most indicators are either national or local scale, and thus not interpretable at a high resolution for the entire country. Here, we include five indicators representing the health domain: the proportion of population having suffered from diseases caused by disasters, the proportion of population having experienced diarrheal disease during disaster periods, lack of drinking water due to disasters, the number of hospital beds, and the number of physicians.

The Min-Max formula is applied to derive an indicator score of Upazila $i$ as follows:

$$Indicator\ Score_i = \frac{x_i - x_{min}}{x_{max} - x_{min}} \qquad (1)$$

where $x_i$ is the original value of the indicator, and $x_{min}$ and $x_{max}$ are the lowest and highest values of the indicator, respectively. Indicator scores range from zero to one, with larger values representing an increase in vulnerability (Table 2). All data is normalized to account for differences in magnitude of units.

### 3.1.2 Vulnerability calculation

In this study, equal-weight and PCA approaches are proposed to calculate $DV$ for the three domains (Table 2). The equal-weight approach applies the addictive model with equal weights for all indicators in a domain as follows:

$$DV_i = \frac{\sum_{k=1}^{n} IS_{i,k}}{n} \qquad (2)$$





where $DV_i$ denotes the domain vulnerability index of Upazila $i$, and $IS_{i,k}$ is $k$th indicator score of Upazila $i$ (here $n$ indicates
the number of indicators in each domain shown in Table 2).

PCA is a common data-driven approach for construction of the Social Vulnerability Index proposed by Cutter et al. (2003).
Specifically, PCA reduces the number of indicators to a smaller number of components that account for a significant portion
of the variances of the indicators. Through grouping highly correlated and similar indicators, principal components (PC) are
formed. Here, varimax rotation is used to create more independence between PCs. Only PCs with eigenvalues > 1 are retained
in order to meet the Kaiser criterion (Kaiser, 1960). The domain vulnerability index for each Upazila is calculated by adding
the scores of all the retained PCs as follows:

$$SHV_i = \frac{DV_{Socio-eonomic_i} + DV_{Health_i} + DV_{Coping\ capacity_i}}{3} \tag{3}$$

Thus, each domain vulnerability contributes equally to the SHV index value. SHV scores are classified into five categories
based on their values: very-low vulnerability (0 to 0.2), low vulnerability (0.2 to 0.4), moderate vulnerability (0.4 to 0.6), high
vulnerability (0.6 to 0.8), and very-high vulnerability (0.8 to 1).

### 3.2   Impact assessment linking with flood forecast and satellite inundation information

Although social vulnerability is a complex function of social, economic, and cultural context, numerical vulnerability estimates
are often presented in terms of fatalities, economic losses, migration, etc. (Rufat et al., 2019; Villagrán de Léon, 2006). One
can imagine that a region classified as highly vulnerable may experience severe impacts from a disaster, poor resilience, slow
recovery, or high rates of a particular action such as displacement or emergency shelter use (Fekete, 2009). However, validation
of social vulnerability is typically challenging due to limited availability and quality of data during/after the disaster period.
Moreover, given the compound characteristic of a composite vulnerability, a comparison of vulnerability with a particular
disaster outcome may be difficult to validate in a traditional sense (Rufat et al., 2019). That withstanding, the objective here
is to develop vulnerability measures for impact assessment, and specifically evaluate its utility for the August 2017 flood
by merging with physical flood hazard information (i.e., flood forecast and satellite inundation) in order to aid in pre- and
post-disaster management practices.

In 2017, after devastating floods in the pre-monsoon period (mid-April) and the monsoon period (early July), the second
monsoon rains began on August 11th, causing intense floods in 42% of the country, including 5 divisions and 32 districts in
the northern, northeastern and central parts of the country, affecting a total of more than 11 million people. According to the
Ministry of Disaster Management and Relief, this flood has been recorded as the worst in the last four decades (FFWC, 2018;
NIRAPAD, 2017b).

First, we estimate the affected population based on flood forecast and satellite inundation maps and spatial population data.
All spatial data is linearly downscaled to a 30m resolution. Flood forecasts are represented as flood depths. The affected
population is assumed to increase linearly from no impact at a flood depth of zero to maximum impact at a flood depth of
3 m. Satellite-based inundation conveys whether or not a grid (30m resolution) is flooded. For flooded grids, we assume the
full population of that grid is affected. We acknowledge that this may result in an overestimation of the affected population,


however explicit flood protection infrastructure data is not available widely. This approach does still capture spatial patterns of affected population.

Post-disaster records were aggregated and reported at the district-level for the August 2017 flood event, therefore we calculate the district-level domain vulnerability by taking the population-weighted average of Upazila-level domain vulnerabilities as follows:

$$DV_j = \frac{\sum_{i=1}^{n}(POP_i \times DV_i)}{POP_j} \tag{4}$$

where POP and DV are the affected population and domain vulnerability of Upazila $i$ and district $j$, respectively. Thus, the district-level DV indicates the average vulnerability of the affected population in each district.

In lieu of evaluating and comparing vulnerability directly with all disaster outcomes, we group the disaster records into four index types, including distress, damage, disruption, and health (Table 3), as utilized by local management agencies and defined in post-disaster reports. Specifically, the distress index includes the percentage of the affected population and the number of deaths, the damage index includes the number of damaged houses and crop land areas, the disruption index includes the number of affected educational institutions and damaged tube-wells, and the health index includes the number of diarrheal and other disease cases (e.g., injury, drowning, RTI, skin, snakebite, etc.) (Table 3).

The variables within each group are normalized and averaged to form a group impact score. Validation is carried out by calculating correlations between developed vulnerability scores and group flood impact scores.

## 4 Results

### 4.1 Relationships between vulnerability indicators

To evaluate cross-correlations, selected indicators (Table 2) are compared at the Upazila or district levels (Figure 2). As necessary, Upazila level indicators are upscaled to compare with district-level indicators using population or household weights.

In general, socio-economic domain indicators are positively correlated with each other. In particular, demographic and built-environment indicators have a significantly positive correlation with most all socio-economic indicators. Within the socio-economic domain, only the ratios of ethnic population and rented houses exhibit some negative correlation with other socio-economic indicators. The socio-economic domain indicators show little correlations with district-level indicators in other domains, which may be due to reduced variability through the upscaling process. In the health domain, the number of hospital beds (number of physicians) show significantly positive (negative) correlations with most socio-economic indicators. However, the percentages of population who suffered from disease or experienced diarrhea from disaster are poorly correlated with other indicators. The coping capacity domain includes indicators that represent historic flood impacts, such as the number of households affected by floods and the number of children not attending school due to disasters. These indicators present positive correlations with indicators in the socio-economic domain, including the percentage of weak population, building materials, electricity, literacy, and education level, which may imply their root causes. The indicator for households with disaster knowledge and perceptions does not correlate well with any other indicators due to its relatively low variability across districts.



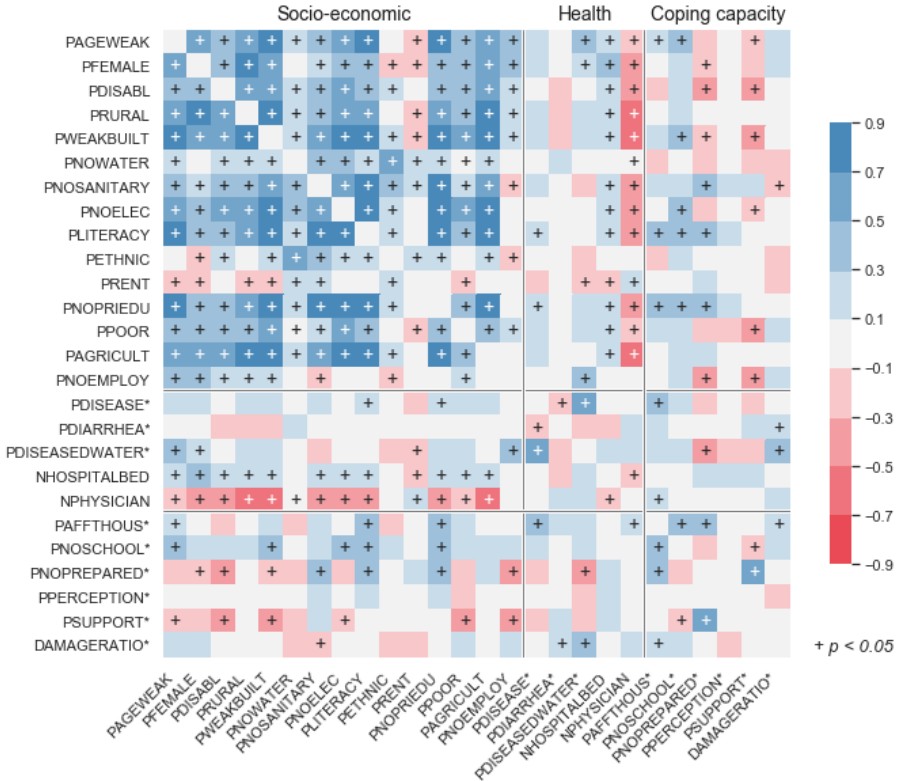

**Figure 2.** Cross-correlation matrix of the selected indicators calculated at Upazila-level unless followed by an asterisk (district-level.) The plus sign indicates a statistically significant correlation (p < 0.05).

### 4.2 Vulnerability assessment

Spatial representation of the DV is determined using the equal weight and PCA approaches for each of the three domains: socio-economic, health, and coping capacity (Figure 3). In the PCA analysis, 3 PCs are included in the socio-economic and coping capacity domains, and 2 PCs are retained in the health domain per the eigenvalue criterion (Figure S2).

Both socio-economic DVs based on the two approaches clearly represent the expected demographic, social, and economic characteristics of major cities, for example lower vulnerability (standard deviation (SD) < -1.0) near the country center (Dhaka; 240 capital city) and the southeast coast (Chittagong; the second largest city) and high vulnerability (SD > 1.0) in the northeast floodplain and sparsely populated southeast region (Figure 3a and 3d.) While the equally weighted socio-economic DV results in relatively high vulnerabilities in the northern and northwestern regions, the PCA-based socio-economic DV exhibits medium vulnerabilities in these regions. Interestingly, the first PC of the socio-economic domain has a very similar pattern with the equally weighted socio-economic DV (r = 0.87), which implies that the equally weighted socio-economic DV reflects the 245 pattern with the largest variance of the variables, however this is modulated by the other two PCs representing a relatively lower vulnerability for these regions (Figure S2a). The PCA-based socio-economic DV also produces exceptionally high

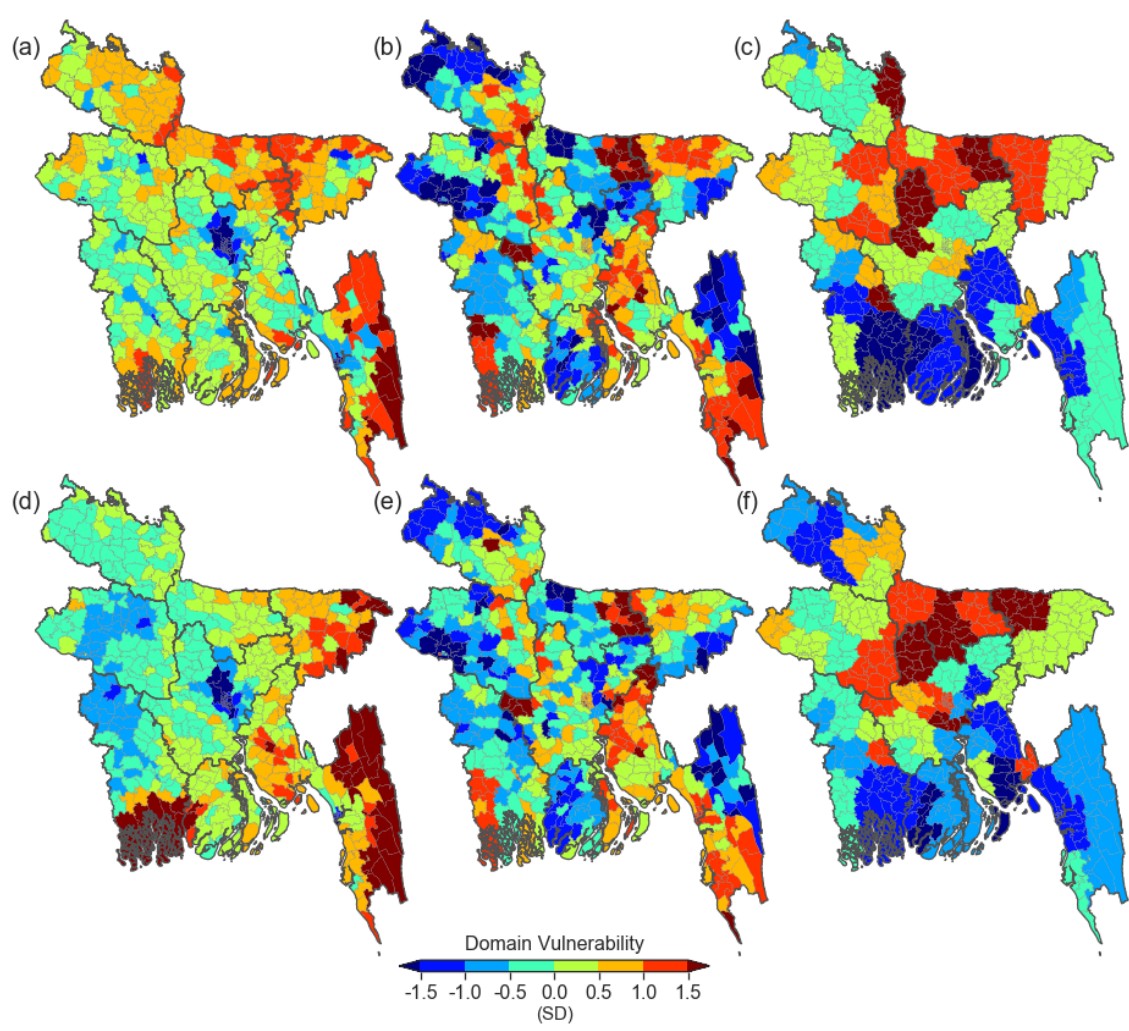

**Figure 3.** The domain vulnerability of (a and d) socio-economic, (b and e) health, and (c and f) coping capacity domains using the equal weight (upper row) and PCA (bottom row) approaches. The unit is the standard deviation (SD) from the mean.

vulnerabilities (SD > 1.5) in the southwestern coastal and southeastern mountain regions due to the second PC pattern (Figures 3 and S2b).

The health DV exhibits very similar patterns between the two approaches (r=0.91). Relatively high health DV are found in
the northeastern floodplains, northwestern riverine areas, and southeastern regions (Figures 3b and 3e). The coping capacity DV also expresses very similar patterns between the two approaches (r = 0.84). The central and northern regions are highly vulnerable, while the southern regions are relatively less vulnerable. The second PC of the coping capacity DV produces the highest correlation (r = 0.58) with the equally weighted coping capacity DV (Figures 3c and S2g). Given equivalent prioritization of all three domains, regions with relatively high vulnerability in all domains tend to have high SHV scores, such
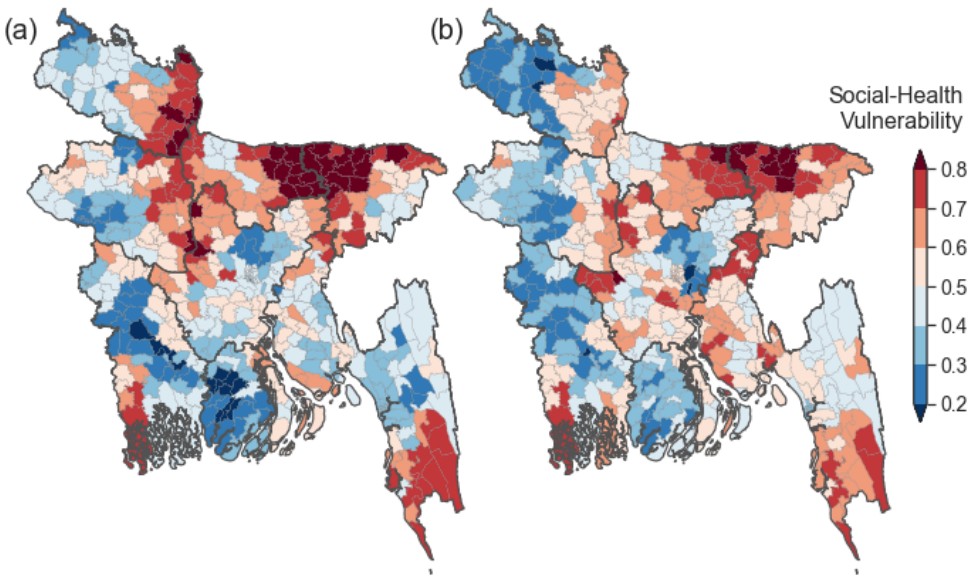

**Figure 4.** The SHV index score estimated by (a) the equal-weight and (b) PCA approaches.

as in the northeastern floodplains (Figure 4). The major difference between the two approaches appears in the northwestern riverine regions; while the equal-weight approach indicates a relatively higher vulnerability ($\geq 0.7$), the PCA approach yields moderate vulnerability ranging from 0.4 to 0.6 (Figure 4). As discussed above, this difference is mainly due to a relatively lower socio-economic DV in the PCA approach (Figure 3d).

     For both approaches, vulnerable zones ($\geq 0.6$) appear proximal to major rivers and tributaries from northwest to central
Bangladesh, and more broadly across low floodplains in the northeast (Haor basin; Figure 4). Although we did not include any physical factors (e.g., proximity to river), this could imply that historical floods have led to the disruption and depreciations of riverine communities. The identification of the Haor region as vulnerable provides confidence in the framework, although not surprising, given that Haor is typically flooded for 7-8 months each year and generally has a low socio-economic status (ACAPS, 2014; Start Network, 2018). High socio-economic, health, and coping capacity DVs (Figure 3) contribute to this
elevated vulnerability (SHV $\geq 0.8$) as measured across the region by both approaches. There are also highly vulnerable zones around the southeastern border between Bangladesh and Myanmar, which are sparsely populated with high socio-economic DVs (ACAPS, 2014) (Figure 3 and 4)

     Low ($\leq 0.4$) and very-low ($\leq 0.2$) SHV scores exist for densely populated areas, such as Dhaka, Chittagong, and south-western regions (Khulna division) where socio-economic, health, and coping capacity DVs are typically low (Figure 3 and
4). Southern coastal regions are often classified as vulnerable regions due to periodical coastal hazards and cyclones, however, both approaches present very low coping capacity DVs ($\leq -1.0$) in these regions (Figure 3). This is because our coping capacity indicators have included existing active disaster management practices and financial supports from agencies in those regions that have resulted in low coping capacity DV and SHV.



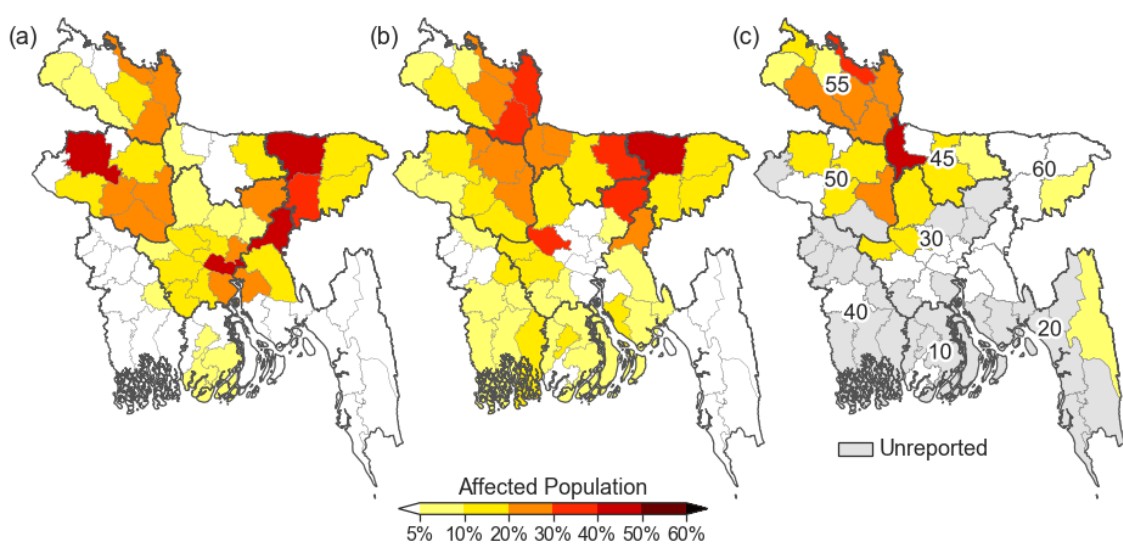

**Figure 5.** Percentage of affected population from (a) flood forecast, (b) satellite inundation, and (c) post-disaster reports. The division codes are Barisal (10), Chittagong (20), Dhaka (30), Khulna (40), Mymensingh (45), Rajshahi (50), Rangpur (55), and Sylhet (60).

On average, considering approaches, half of the country (45%) is classified as moderately vulnerable ($0.4 \leq$ SHV $< 0.6$),
with the remaining 55% split between high vulnerability zones (SHV $\geq 0.6$), including 42 million people or 26% of the population, and low vulnerability zones (SHV $< 0.4$), with 46 million people or 29% of the population. As proposed in the framework (Figure 1), DV and SHV can also be merged with physical flood information to assess predictability of flood impacts. However, identifying highly vulnerable zones based solely on indicators is also informative for government and relief agencies to enhance the resilience of these regions through long-term management practices.

**4.3  Impact assessment**

For the August 2017 event, flood forecast and satellite inundation estimates indicate that 16.8 (10.6%) and 15.3 (9.7%) million people nationally were impacted from flood inundation, respectively. Post-disaster reports claim 9.2 million (5.8%) of the population was impacted (Figure 5). This overestimation is likely attributable to the simplified approaches and insufficient data quality. For example, the two approaches adopted here do not consider the level of flood protection (e.g., embankment,
levee, early warning, etc.) but rather assume all regions have equivalent protection and management. Furthermore, the current flood forecasts and satellite inundation information do not provide specific physical flood properties, such as duration of the flood, which is a key factor in increasing flood impacts, as indicated in the post-flood reports. Geographical contexts may also contribute to this discrepancy. For example, both forecasts and satellite information estimate a high number of affected people in the northeastern floodplain (i.e., Haor region), whereas a relatively low percentage of affected population is reported (Figure
5). This region is known to be highly vulnerable to flooding, but home styles and small households (lowest population density in Bangladesh) are well adapted to regular monsoon floods (ACAPS, 2014). At the Upazila-scale (Figure S3) impacted population



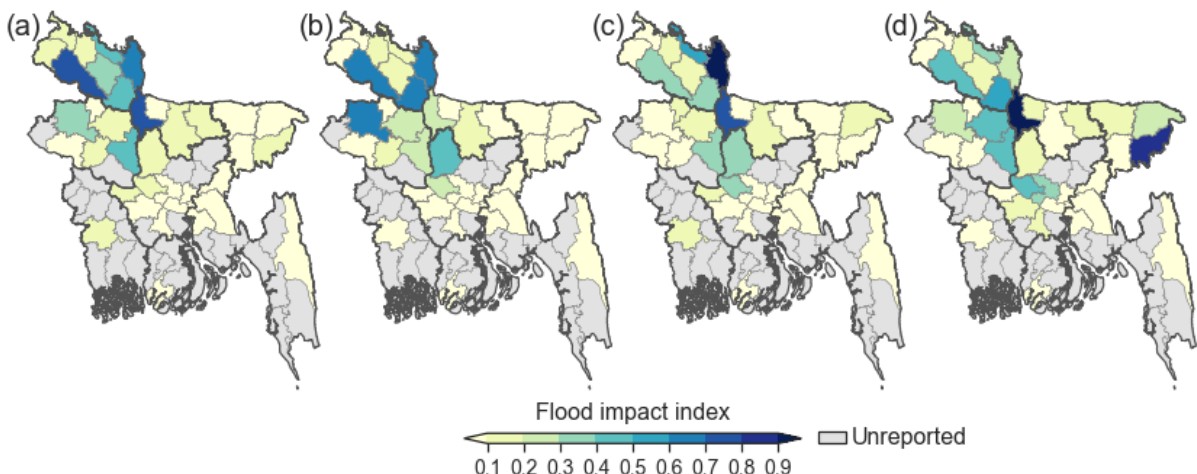

**Figure 6.** Impact index maps of the August 2017 flood event for (a) distress, (b) damage, (c) disruption, and (d) health.

estimates from these two sources differ. For example, relatively highly affected populations appear near major rivers based on the forecast, such as the Jamuna (northwest) and Meghna (northeast) rivers, however the satellite information illustrates highly affected population more broadly around riverine areas. Spatially, satellite-based estimates correlate better with the reported affected population (r = 0.6) than the forecast-based result (r = 0.1). Specifically, satellite inundation captures severely flooded regions in the northwestern, as reported.

Four indices of flood impact records (distress, damage, disruption and health) are normalized and compared to the tailored district level SHVs and DVs (Figures 6 and S4; Table 4). According to post-flood reports, the August 2017 event had a significant impact on the northwestern regions (Rangpur, Rajshahi, and Mymensingh divisions). Generally, the equal-weight approach produces higher correlations than the PCA approach (Figure S4). This is mainly attributable to the relatively low socio-economic DV in the PCA approach for the northwest region. The forecast and satellite-based DVs correlate similarly with the four indices from the two approaches, although the forecast-based are marginally higher, and correlations with equal weights are notably higher than for the PCA approach (Table 4). Again, the moderate vulnerability of the PCA approach on the northwestern regions substantially depreciates its correlations with overall flood impact indices (Figures 4 and 6).

Specifically, the forecast-based socio-economic DV spatially correlates well with the equal weights approach indices, statistically significantly capturing distress (r = 0.38) and disruption (r = 0.3) impact indices. For the same comparison, the coping capacity DV also produces statistically significant correlations with disruption (r = 0.34) and health (r = 0.31) impact indices. Surprisingly, the health DV demonstrates a low correlation with the health impact index, which consists of diarrheal and other disease incidents. Given that the causes of disease outbreaks are quite complex (e.g., current vaccines and medical status) and often do not have a simple relationship with hazard (Shahid, 2010), this reiterates that considering a capability to prepare/manage natural disasters may provide a better indication of the likelihood of flood-induced health impacts and epidemics as discussed by previous studies (Hashizume et al., 2008; Kunii et al., 2002; Schwartz et al., 2006).





Overall, the forecast-based SHV is statistically significantly correlated with all types of flood impact indices (Table 4). This could play a critical role in disaster management by indicating comprehensive impacts across multiple sectors.

## 5 Conclusions and Discussions

This study presents development of flood-induced social and health vulnerability measures and evaluates the predictability of flood impacts by linking vulnerability measures to flood forecast and satellite inundation information. Vulnerability domains and indicators are developed at the Upazila level for Bangladesh based on literature review, and three domain vulnerability (DV) categories (socio-economic, health, and coping capacity) and a composite social-health vulnerability (SHV) metric are spatially constructed using both equal weight and PCA approaches (Figure 3 and 4). The DVs and SHV are scaled up to the district level conditioned on affected population weights estimated from flood forecast and satellite inundation information. The predictability of flood impacts is assessed by comparing the tailored vulnerability measures with observed flood impact indices (distress, damage, disruption, and health) aggregated from post-disaster reports on the August 2017 flood. Such an evaluation has not been previously undertaken.

The proposed approach shows promising results. First, we find highly (SHV $\geq 0.6$) and very-highly (SHV $\geq 0.8$) vulnerable zones near the northwest riverine areas, northeast floodplains, and southwest region covering 42 million people (26% of total population); most indicators illustrate consistently high vulnerability levels (Figure 4). A spatial discrepancy in SHV between the equal weight and PCA approaches in the northwest riverine regions is evident, however, mainly attributable to the socio-economic DV.

The affected population by the August 2017 flood event is estimated using flood forecast and satellite inundation information (Figure 5). Although both sources overestimate the affected population due to a lack of the information, such as flood protection/management and duration of flood, the satellite-based information exhibits a fairly consistent spatial pattern with the reported population (r =0.6). Given that the socio-economic DV is strongly correlated with the distress impact index, which includes the number of affected people and deaths (Table 4), the inclusion of a socio-economic DV to represent the level of overall flood protection and management is warranted. For this analysis, the equal weight approach has a stronger relationship with flood impact indices than the PCA approach (Table 4). Specifically, the socio-economic DV reflects the distress impact, and the coping capacity DV captures disruption and health impacts. This suggests that thematic vulnerability can play an important role in contextualizing flood impacts.

Although the health vulnerability measure consists of indicators related to previous disease incidents, lack of drinking water, hospital capacity, and health workforce, it does not reflect well the observed health impact. However, the coping capacity DV does, suggesting that low resilience and lack of recovery mechanisms may better represent the potential for flood-induced disease outbreaks. Overall, the forecast-based SHV index exhibits a significant relationship with all flood impact indices, demonstrating its usefulness in identifying vulnerable regions. Satellite-based vulnerability measures are also promising, which may be especially useful for countries with limited capacity to build flood forecast systems. Whereas these tailored vulnerability measures can support pre- and post-flood disaster management activities, the original vulnerability measures based solely on





the indicators (Figure 4) can support identification of areas requiring long-term investment and management plans to reduce particular aspects of vulnerability. Additionally, given that current post-disaster reports include only disaster-related statistics at the district level (BBS, 2016; NIRAPAD, 2017a), the Upazila-level vulnerability measures developed in this study can provide more specific and useful information as thematic or comprehensive vulnerability and resiliency measures.

For the first time, we incorporate predictive information into the vulnerability and validate it with a recent catastrophic flood event. Compared to conventional and existing approaches, the vulnerability measures developed here have unique advantages and contributions. First, given the lead-time of flood forecast, the predicted vulnerability can dynamically anticipate flood impacts and actively support pre- and post-disaster management rather than being used as static supplementary data. We also demonstrate, through a validation, that the thematic vulnerability can better estimate a particular aspect of flood impacts.

This can potentially facilitate tailored management actions, such as prioritizing different resources (e.g., foods, cash, medical supplies, volunteers, etc.) for the given location. In order to enhance the quality of this approach, the validation process can be updated and improved with additional data, indicators, and flood records across the country to enhance management practices. Especially, more accurate post-disaster impact records with diverse variables at the local level (e.g., Upazila scale) may improve future vulnerability and risk assessments and impacts prediction. Flood forecasts have clear value, however producing local

scale information may pose challenges in countries with limited resources; existing global scale forecasts may be able to fill this role and should be evaluated (Alfieri et al., 2013; Emerton et al., 2018). Understanding the prospects for extending forecast lead-times is also warranted, and may facilitate more proactive disaster management practices (Coughlan de Perez et al., 2015). Finally, integrating more physical flood information and models to estimate the affected population may enhance flood impact predictions.

The proposed approach is transferable and easily adapted to different countries to assess vulnerabilities. Integrating this approach systematically with a flood forecast system, such as a web-based online tool, may be of further value to international and local disaster managers. However, it should be noted that the quality of this approach depends on the quality and availability of data, particularly for demographic and socio-economic elements requiring a sub-national level census. Overall, this study provides groundwork for the development of a multi-sectoral (flood and health) risk warning system. Actionable flood and

health risk predictions can radically improve existing disaster management practices of NGOs and other private and public organizations and save lives and resources by providing advanced preparedness and response strategies.

*Data availability.* The original data of vulnerability indicators are publicly available (see Section 2.1). The processed vulnerability indicators can be requested from the corresponding author.

*Author contributions.* The paper and its methodology were conceptualized and developed by DL and PB; the data processing, analyses, and

visualization were carried out by DL. The original draft was prepared by DL and PB; further reviewing and editing was carried out by DL, PB, JP, and HA.



*Competing interests.* The authors declare that they have no conflict of interest.

*Acknowledgements.* The lead author would like to thank Mr. Arifuzzaman Bhuyan at FFWC for providing flood forecast data.

*Financial support.* This research is partially funded by the Global Health Institute (GHI) of the University of Wisconsin-Madison, the
Wisconsin Alumni Research Foundation (WARF) – UW2020, and NASA project NNX17AC50G.



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



Natural Hazards
and Earth System
Sciences

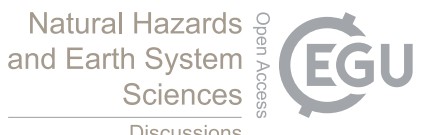

Discussions

**Table 1.** Report and studies of spatial social vulnerability to flood disasters in Bangladesh.

| Name | Hazard | Vulnerability assessment method | Study area (scale) | Data sources |
|---|---|---|---|---|
| Composite vulnerability index to flood (Dewan, 2013) | Flood | Selection method: Literature review and data availability. Domains and indicators: Physical (4), social (9), and coping capacity (5). Vulnerability model: Addictive model with weights from AHP analysis for each domain. The CVI is calculated by adding domain vulnerabilities equally. | Dhaka city (Community level, Gridded) | Population census of 2001, Household Income and Expenditure Survey of 2005, geospatial data, other studies |
| Climate hazard vulnerability (Islam et al., 2013) | Flood, drought, tidal surge | Selection method: Adopted from Patnaik and Narayanan (2009). Domains and indicators: Agricultural (5), climatic (2), occupational (4), demographic (2), and geological (2) variables. Vulnerability model: Addictive model with equal weights | All districts across the country (District level) | Statistical yearbook of Bangladesh (1974-2006) |
| Socioeconomic vulnerability index (Ahsan and Warner, 2014) | Climate change (Coastal) | Selection method: Participatory rural appraisal (PAR) model. Domains and indicators: Demographic (5), social (5), economic (6), physical (5), and exposure (6). Vulnerability model: Addictive model with weights obtained from workshop | 7 unions in Koyra Upazila in the southwestern coastal Bangladesh (Union level) | Rapid rural appraisal, household survey, workshop |
| Spatial vulnerability to flood (Roy and Blaschke, 2015) | Flood | Selection method: Literature review, data availability, consulting with experts and community members. Domains and indicators: Adopted from (Kienberger et al., 2009), Sensitivity (35) and Coping capacity (9). Vulnerability model: Addictive model with weights from AHP analysis | 10 unions in Dacope Upazila in the southwestern coastal Bangladesh (Gridded) | Household survey, geospatial data |
| Flood vulnerability in flood risk assessment (Gain et al., 2015) | Flood | Selection method: Literature review. Domains and indicators: Adaptive capacity (4), coping capacity (2), and susceptibility (3). ulnerability model: Addictive model with equal weights | Eastern part of Dhaka city (Gridded) | Population census of 2011, geospatial data, other studies |


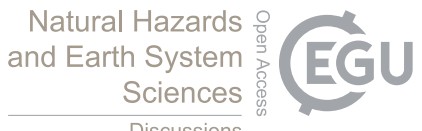

**Table 1.** Report and studies of spatial social vulnerability to flood disasters in Bangladesh (continued).

| Name | Hazard | Vulnerability assessment method | Study area (scale) | Data sources |
|---|---|---|---|---|
| Exposure, vulnerability, and risk assessment for natural disasters (DDM, 2017) | Flood, storm surge, landslide, drought, earthquake, tsunami | Selection method: Not mentioned. Domains and indicators: Exposure assessment is applied to each variable in population (7), housing (1), livelihood (2), critical facilities (4), infrastructures (5) domains. Vulnerability model: Individual variables are analyzed for exposure assessment to each hazard. Multi-hazard vulnerability and risk scores are calculated based on pre-defined standards (e.g., % of affected population and households) | All districts (Upazilas) across the country (Upazila level, Gridded) | Population census of 2011, geospatial data |
| Social vulnerability to flood (Netherlands Red Cross, 2017) | Flood | Selection method: Not mentioned. Domains and indicators: Poverty incidence (1) and deprivation index (1). The deprivation index is a composite index calculated from 21 socio-economic variables using PCA analysis (Mahhzab, 2015). Vulnerability model: The geomean value of two indices. | National (Union level with district level data) | Poverty index is from WorldPop |
| Flood vulnerability (Hoque et al., 2019) | Flood | Selection method: Literature review and data availability. Domains and indicators: Physical (5), social (8), and coping capacity (3). Vulnerability model: Physical vulnerability × social vulnerability / coping capacity with weights from AHP analysis | 11 unions in Kalapara Upazila in the central coastal Bangladesh (Gridded) | Population census of 2011, geospatial data, fieldwork |
| Social vulnerability (Rabby et al., 2019) | Coastal | Selection method: Adopted from Cutter et al. (2003) to consider coastal Bangladesh. Domains and indicators: 9 principal components from 27 socio-economic variables. Vulnerability model: Addictive model of 9 principal components with no weights | 19 districts (1521 unions) of the coastal Bangladesh (Union level) | Population census of 2011 |

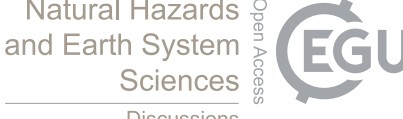

Natural Hazards and Earth System Sciences Discussions — Open Access — EGU

**Table 2.** Selected indicators for vulnerability assessment.

| Domain | | Name | Description | Relationship | Scale | Source |
|---|---|---|---|---|---|---|
| Socio-economic | Demographic | PAGEWEAK | % of weak population (age below 5 or above 65 years) | + | Upazila | BBS (2015) |
| | | PFEMALE | % of woman | + | Upazila | BBS (2015) |
| | | PDISABL | % of population with any sort of disability | + | Upazila | BBS (2015) |
| | Built-environment | PRURAL | % of households in rural areas | + | Upazila | BBS (2015) |
| | | PWEAKBUILT | % of households with weak materials | + | Upazila | BBS (2015) |
| | | PNOWATER | % of households without public water supply | + | Upazila | BBS (2015) |
| | | PNOSANITARY | % of households without sanitary facilities | + | Upazila | BBS (2015) |
| | | PNOELEC | % of households without electricity | + | Upazila | BBS (2015) |
| | Social | PLITERACY | % of population who cannot read and write | + | Upazila | BBS (2015) |
| | | PETHNIC | % of ethnic population | + | Upazila | BBS (2015) |
| | | PRENT | % of rented houses | + | Upazila | BBS (2015) |
| | | PNOPRIEDU | % of population without primary education | + | Upazila | BBS (2015) |
| | Economic | PPOOR | % of population below the upper poverty line | + | Upazila | BBS (2010) |
| | | PAGRICULT | % of population engaged in agriculture work | + | Upazila | BBS (2015) |
| | | PNOEMPLOY | % of population without employment | + | Upazila | BBS (2015) |
| Health | | PDISEASE | % of population who has suffered from disease due to disasters | + | District | BBS (2016) |
| | | PDIARRHEA | % of population experienced diarrhea as a main disease due to natural disaster | + | District | BBS (2016) |
| | | PDISEASEDWATER | % of households with disease due to insufficient drinking water during/after disaster period | + | District | BBS (2016) |
| | | NHOSPITALBED | Number of hospital beds per 1,000 people | - | Upazila | MHFW (2020) |
| | | NPHYSICIAN | Number of physicians per 10,000 people | - | Upazila | DGHS (2020) |
| Coping capacity | | PAFFTHOUS | % of households affected by floods | + | District | BBS (2016) |
| | | PNOSCHOOL | % of children did not attend to school due to disasters | + | District | BBS (2016) |
| | | PNOPREPARED | % of households have not taken disaster preparedness activities | + | District | BBS (2016) |
| | | PPERCEPTION | % of households with knowledge and perception about disaster | - | District | BBS (2016) |
| | | PSUPPORT | % of households received financial support from agencies during/after disaster | - | District | BBS (2016) |
| | | DAMAGERATIO | Ratio of total damage/loss to total income (in USD) | + | District | BBS (2016) |





**Table 3.** Flood impact indices and variables considered for the August 2017 flood event.

| Index | Description | Source |
|---|---|---|
| Distress | Percentage of affected population | NIRAPAD (2017a) |
| | Number of deaths | NIRAPAD (2017a) |
| Damage | Number of damaged houses | NIRAPAD (2017a) |
| | Areas of damaged cropland | NIRAPAD (2017a) |
| Disruption | Number of affected educational institutions | NIRAPAD (2017a) |
| | Number of damaged tube-wells | NIRAPAD (2017a) |
| Health | Number of diarrhea cases | DGHS (2020) |
| | Number of other health outcome cases (injury, drowning, RTI, skin and eye disease, and snake bite) | DGHS (2020) |

**Table 4.** Correlation between SHV and domain vulnerability and observed flood impact indices.

| Vulnerability type | | Equal weight | | | | PCA | | | |
|---|---|---|---|---|---|---|---|---|---|
| | | Distress | Damage | Disruption | Health | Distress | Damage | Disruption | Health |
| Forecast-based | SHV | 0.28* | 0.28* | 0.33*** | 0.32* | 0.01 | 0.01 | 0.1 | 0.13 |
| | Socio-econ | 0.38*** | 0.2 | 0.3* | 0.26 | -0.1 | -0.21 | -0.09 | 0.06 |
| | Health | 0.07 | 0.15 | 0.11 | 0.14 | 0.04 | 0.11 | 0.07 | 0.03 |
| | Coping capacity | 0.19 | 0.27 | 0.34*** | 0.31* | 0.06 | 0.08 | 0.21 | 0.19 |
| Satellite-based | SHV | 0.26 | 0.27 | 0.32* | 0.29* | -0.05 | -0.05 | 0.04 | 0.08 |
| | Socio-econ | 0.35*** | 0.17 | 0.26 | 0.17 | -0.2 | -0.3* | -0.2 | -0.06 |
| | Health | 0.06 | 0.14 | 0.1 | 0.14 | 0.02 | 0.09 | 0.05 | 0.03 |
| | Coping capacity | 0.18 | 0.26 | 0.33*** | 0.29* | 0.05 | 0.06 | 0.19 | 0.17 |

*** $p < 0.01$, ** $p < 0.05$, * $p < 0.1$