# Peer review of "Predicting social and health vulnerability to floods in Bangladesh"

_Natural Hazards and Earth System Sciences, 2020_

## Referee Comment (RC1) · Anonymous Referee #1 · 27 Dec 2020

Generally, the manuscript is very well written and provides an innovative and simple approach to assess vulnerability. The structure of the paper is clear. Data and methods are well explained, and possible lacks are explained in detail. The title is appropriate and reflects what is reported in the document. I have few minor revisions to propose. In my opinion, the authors should improve and clarify some terms used in the manuscript such as "indicator", "index", and "score". I suggested specifying the meaning of these terms referring to the existing bibliography. The figures are for the most part clear and well organized but cartographic elements such as the North and scale bar should be added at least for the first one, as well as the presence of UTM coordinates on grids.

The references are relevant and sufficient. However, some statements need to be supported by references, in those cases indicated in the revision, they must be added.

Here below specific points and minor issues.

Abstract The abstract is clear and short. I do not have any suggestions

Introduction For this chapter, I have just one suggestion. The authors produce a good explanation of the work and the importance of it in a Bangladesh contest. Moreover, sentences are followed by proper references. My suggestion regards the absence of a location map. For the readers, it would be very useful to have a location map reporting the altitude and some locations mentioned in the manuscript ("Dhaka", "Chittagong", "Haor basin"). This can guide the reader to focus on the more interesting and significant areas of the work itself.

Line 60: "social-health vulnerability (SHV) indicator". Term "indicator is correct? In figure 4 the authors use SHV "index". Please clarify the question of the terms as explained in the general comments.

Data The data sources are well explained. I do not have any suggestions.

Methods Line 139: Please add Index or indicator to "SHV" Line 140 to 142. In my opinion, this sentence has to be reformulated and clarified with more detail. Line 144: "Previous studies". In my opinion relevant references are needed Line 148: A citation is needed after the word "hazard" Line 150: Some citations are needed after the words "etc"

Results The presentation of the results is satisfactory, although in some sentences are too simplified. For example, at section 4.3 the sentence between line 282 and 285 should be better explained and developed. The overestimation due to insufficient data quality should be verified more in detail, overall if the authors declare that the proposed approach is "transferable and easily adapted to different countries". The data quality is not analysed with enough detail. I suggest to spend some lines to explain the possible consequences of the data quality or missing data on the proposed approach. Discussion and conclusion In my opinion this chapter needs to be rearranged. Discussion

and conclusion should be divided in two different chapters. Furthermore, in the discussion chapter, the authors should compare their own approach with other international published approaches. This suggestion is made in order to highlight the quality of the approach shown and which are the advantages compared with other existing approaches. Moreover the sentence "The proposed approach is transferable and easily adapted to different countries to assess vulnerabilities ", is very strong. This approach is used just for a single case of study and in a specific country. I think that this sentence should be reformulated due to the fact that the method has different constrains and should be clear in what conditions of data and knowledge it could applied.

---

## Author Comment (AC1) · 5 Jan 2021

Authors' replies are in blue color and revised sentences are in italics.

Anonymous Referee #1

Generally, the manuscript is very well written and provides an innovative and simple approach to assess vulnerability. The structure of the paper is clear. Data and methods are well explained, and possible lacks are explained in detail. The title is appropriate and reflects what is reported in the document. I have few minor revisions to propose. In my opinion, the authors should improve and clarify some terms used in the manuscript such as "indicator", "index", and "score". I suggested specifying the meaning of these terms referring to the existing bibliography.

1) We thank the anonymous reviewer for the positive comments and further critical comments that we believe have enhanced the overall quality of the manuscript. As you suggested, we now better clarify the terms "indicator", "index", and "score". Specifically, we have provided the following sentence at Line 68 (at the beginning of Data section):

   *In this study, as is generally adopted in the literature, an indicator represents an individual variable, an index represents a composite vulnerability, and a score indicates the value of an indicator or index (Birkmann, 2006; Fekete, 2009).*

The figures are for the most part clear and well organized but cartographic elements such as the North and scale bar should be added at least for the first one, as well as the presence of UTM coordinates on grids.

2) We agree with the reviewer. We have added a location map showing flood forecast and inundation with key regions explained in the manuscript (e.g., Dhaka, Chittagong, and Haor basin). Please find the Figure 1 below.

The references are relevant and sufficient. However, some statements need to be supported by references, in those cases indicated in the revision, they must be added.

3) We have added additional references. Please find our specific answers to your comments below.

Here below specific points and minor issues.

Abstract
The abstract is clear and short. I do not have any suggestions

Introduction
For this chapter, I have just one suggestion. The authors produce a good explanation of the work and the importance of it in a Bangladesh contest. Moreover, sentences are followed by proper references. My suggestion regards the absence of a location map. For the readers, it

would be very useful to have a location map reporting the altitude and some locations mentioned in the manuscript ("Dhaka", "Chittagong", "Haor basin"). This can guide the reader to focus on the more interesting and significant areas of the work itself.

4) Thanks for your comment. We have provided a location map (Figure 1).

Line 60: "social-health vulnerability (SHV) indicator". Term "indicator is correct? In figure 4 the authors use SHV "index". Please clarify the question of the terms as explained in the general comments.

5) We have corrected this to read: "social-health vulnerability (SHV) index".

Data
The data sources are well explained. I do not have any suggestions.

Methods
Line 139: Please add Index or indicator to "SHV"

6) We have changed "SHV" to "SHV index" throughout the manuscript.

Line 140 to 142. In my opinion, this sentence has to be reformulated and clarified with more detail.

7) To clarify, we have changed Lines 140-142 to:

*Some studies of vulnerability assessments to flood include physical indicators (e.g., floodplain area, low elevation, proximity to river, etc.) in their composite vulnerability index to reflect flood susceptibility or risk (Dewan, 2013; Islam et al., 2013; Roy and Blaschke, 2015, Hoque et al., 2019). In this study, however, the SHV index precludes those physical indicators, as flood hazard information (i.e., flood inundation) will be linked later through the impact assessment.*

Line 144: "Previous studies". In my opinion relevant references are needed

8) Thanks for this comment. We have added references to Lines 142-144:

*Instead, we include a health domain uniquely reflecting flood-induced health risk that has rarely been considered in previous studies (Ahsan and Warner 2014, Gain et al., 2015, Rabby et al., 2019) (Table 1).*

Line 148: A citation is needed after the word "hazard"

9) We have added references to Line 148:

*Comparatively, the coping capacity domain represents the ability to cope with or adapt to the hazard (Birkmann, 2006; Villagrán de Léon, 2006).*

Line 150: Some citations are needed after the words "etc"

10) We have added references to Line 150:

*In the literature, coping capacity indicators are surveyed for the local region, or proxy data from the census are used, such as households with communication devices and vehicles, literacy rates, education levels, etc (Dewan 2015; Roy and Blaschke, 2015).*

Results
The presentation of the results is satisfactory, although in some sentences are too simplified. For example, at section 4.3 the sentence between line 282 and 285 should be better explained and developed. The overestimation due to insufficient data quality should be verified more in detail, overall if the authors declare that the proposed approach is "transferable and easily adapted to different countries". The data quality is not analysed with enough detail. I suggest to spend some lines to explain the possible consequences of the data quality or missing data on the proposed approach.

11) Thanks for pointing this out. To clarify, Lines 282-287 have been changed to:

*This overestimation of the affected population is likely attributable to the simplified approaches and a lack of data on flood management and properties. For example, the two approaches adopted here do not consider the level of flood protection (e.g., embankment, levee, early warning, etc.) but rather assume that all regions have an equivalent level of protection and management. Furthermore, the current flood forecasts and satellite inundation information do not provide specific physical flood properties, such as duration of the flood, which is a key factor in increasing flood impacts, as indicated in the post-flood reports.*

12) Also, Line 365 has been changed to:

*Although this study includes Bangladesh as a single case study country, the proposed approach may be transferable to different countries where sufficient data are available. Specifically, given that this approach focuses on social and health vulnerability, the demographic and socio-economic components of vulnerability require at least sub-national level census data. In addition, data on observed flood impacts (i.e., post-disaster reports) are required to validate this approach at other locations.*

Discussion and conclusion
In my opinion this chapter needs to be rearranged. Discussion and conclusion should be divided in two different chapters. Furthermore, in the discussion chapter, the authors should compare their own approach with other international published approaches. This suggestion is made in

order to highlight the quality of the approach shown and which are the advantages compared with other existing approaches.

13) Thanks for your comment. We note that no international published approaches have assessed vulnerability for all of Bangladesh at this resolution. Instead, we have highlighted the new contributions of our approach in comparison with conventional approaches. To clarify, Lines 351-356 have been changed to:

*Compared to conventional and existing approaches, the approach and vulnerability measures developed here have unique advantages and contributions. First, our approach covers all of Bangladesh at a high resolution (Upazila). While local studies provide more specific analyses (e.g., key vulnerability factors within a city), our scale and resolution can support national-level assessment and management when massive monsoon floods affect most of the country. Second, given the lead-time of flood forecasts, the predicted vulnerability can dynamically anticipate flood impacts and actively support pre- and post-disaster management rather than being applied as static vulnerability data as conventional approaches provide. We also demonstrate, through a validation, that the thematic (domain) vulnerability can better estimate a particular aspect of flood impacts. This can potentially facilitate tailored management actions, such as prioritizing different resources (e.g., foods, cash, medical supplies, volunteers, etc.), for the given location.*

14) Also, we have separated and added a Conclusion section as follows:

*6 Conclusions*

*In this study, we assess three domain vulnerabilities and a composite social-health vulnerability for all of Bangladesh. Results indicate that vulnerable zones exist in the northwest riverine areas, northeast floodplains, and southwest region, potentially affecting 42 million people (26% of total population). Then, for the first time, we incorporate predictive information (flood forecast and satellite inundation) into vulnerability and validate it with the recent catastrophic August 2017 flood event. Our findings suggest that the both forecast and satellite-based vulnerabilities can better inform observed flood impacts.*

*Compared to conventional and existing approaches, the approach and vulnerability measures developed here have unique advantages and contributions. First, our approach covers all of Bangladesh at a high resolution (Upazila). While local studies provide more specific analyses (e.g., key vulnerability factors within a city), our scale and resolution can support national-level assessment and management when massive monsoon floods affect most of the country. Second, given the lead-time of flood forecasts, the predicted vulnerability can dynamically anticipate flood impacts and actively support pre- and post-disaster management rather than being applied as static vulnerability data as conventional approaches provide. We also demonstrate, through a validation, that the thematic (domain) vulnerability can better estimate a particular aspect*

*of flood impacts. This can potentially facilitate tailored management actions, such as prioritizing different resources (e.g., foods, cash, medical supplies, volunteers, etc.), for the given location.*

*In order to enhance the quality of this approach, the validation process can be updated and improved with additional data, indicators, and flood records across the country to enhance management practices. Especially, more accurate post-disaster impact records with diverse variables at the local level (e.g., Upazila scale) may improve future vulnerability and risk assessments and impacts prediction. Flood forecasts have clear value, however producing local scale information may pose challenges in countries with limited resources; existing global scale forecasts may be able to fill this role and should be evaluated (Alfieri et al., 2013; Emerton et al., 2018). Understanding the prospects for extending forecast lead-times is also warranted, and may facilitate more proactive disaster management practices (Coughlan de Perez et al., 2015). Finally, integrating more physical flood information and models to estimate the affected population may enhance flood impact predictions.*

*Although this study includes Bangladesh as a single case study country, the proposed approach may be transferable to different countries where sufficient data are available. Specifically, given that this approach focuses on social and health vulnerability, the demographic and socio-economic components of vulnerability require at least sub-national level census data. In addition, data on observed flood impacts (i.e., post-disaster reports) are required to validate this approach at other locations. Integrating this approach systematically with a flood forecast system, such as a web-based online tool, may be of further value to international and local disaster managers. Overall, this study provides groundwork for the development of a multi-sectoral (flood and health) risk warning system. Actionable flood and health risk predictions can radically improve existing disaster management practices of NGOs and other private and public organizations and save lives and resources by providing advanced preparedness and response strategies.*

Moreover the sentence "The proposed approach is transferable and easily adapted to different countries to assess vulnerabilities ", is very strong. This approach is used just for a single case of study and in a specific country. I think that this sentence should be reformulated due to the fact that the method has different constrains and should be clear in what conditions of data and knowledge it could applied.

15) We have changed this sentence. Please refer to our reply #12.


[Figure]

Figure 1. (left) FFWC's flood forecast issued on Aug-16, 2017 and (right) Sentinel-1 based satellite flood inundation for the August 2017 flood event. The borderline represents the boundary of divisions.

[revised manuscript text omitted]

**2.3 Flood impact records**

The Global Shelter Cluster has aggregated relevant post-disaster reports and data for the August 2017 flood event in Bangladesh through government agencies and international relief organizations (https://www.sheltercluster.org/response/bangladesh-monsoon-floods-2017). Specifically, we leverage the 72-hour Rapid Assessment report published August 21st, the flood damage data reported on September 3rd by the DDM and Natural Disaster Response Coordination Group, and monthly hazard incident report from the Network for Information, Response and Preparedness Activities on Disaster (NIRAPAD) (NIRAPAD, 2017b). The DGHS reported health outcomes from the August 2017 flood collected between July to September. From this, we extract the number of diarrheal incidents and other adverse health outcomes, including incidents of respiratory tract infections (RTI), eye and skin diseases, snake bites, drowning, and other injuries.

**3 Methods**

Spatially explicit vulnerability and risk maps can support decision-makers by enhancing their ability to take appropriate actions. However, vulnerability assessment is complicated by environmental, social, economic, and political patterns of societies. To

[Figure]

**Figure 1.** (left) FFWC's flood forecast issued on Aug-16, 2017 and (right) Sentinel-1 based satellite flood inundation for the August 2017 flood event. The borderline represents the boundary of divisions.

date, no standard model or methodology exists to guide spatial vulnerability assessments for natural disasters, although the number of related studies is rapidly increasing (Villagrán de Léon, 2008; Ward et al., 2020). In this study, we select socio-economic, health, and coping capacity vulnerability domains consisting of 26 indicators based on the literature, availability of data, and their vulnerability influences. The domain-level vulnerability (DV) index is estimated using two approaches of vulnerability calculation, namely equal-weights and Principal Component Analysis (PCA); social-health vulnerability (SHV) index is measured using equal weights. The flood forecast and satellite inundation information are applied to estimate affected population during the August 2017 flood event. Finally, vulnerability of the affected population is assessed and validated against records of post-disaster flood impacts. The overall framework of this study is illustrated in Figure 2.

**3.1 Social-health vulnerability assessment**

**3.1.1 Domain and Indicators selection**

Previously, assessments of spatial vulnerability conditioned on socio-economic factors have been conducted for a number of regions of Bangladesh (Ahsan and Warner, 2014; Dewan, 2013; Gain et al., 2015; Hoque et al., 2019; Rabby et al., 2019; Roy

[Figure]

**Figure 2.** Vulnerability and impact assessment framework.

and Blaschke, 2015) and more broadly for the entire country (DDM, 2017; Islam et al., 2013). Method of assessment, indicators, study area, scale, and data are summarized in Table 1. These studies typically select vulnerability domains and indicators based on the context of the target disaster and study area or from a pre-defined approach in the literature. In previous studies, the

135 domains include socio-economic, adaptive or coping capacity, and unique exposure or hazard domains, such as agricultural, physical (climate, flood, or coastal hazard), and infrastructure. For vulnerability models, an addictive model (equal weights) or analytic hierarchy process analysis (AHP) (custom weights from stakeholder engagement or expert opinion (Saaty and Vargas, 2012)) are most common. A PCA analysis (e.g. Cutter et al. (2003)), is also frequently employed to identify dominant spatial patterns and to generate a composite vulnerability. The majority of studies adopt the equal weights approach such that each

140 domain contributes equally to the composite vulnerability  index.

In this study, the SHV index includes 26 indicators  along with three indicator domains: socio-economic (15 indicators), health (5 indicators), and coping capacity (6 indicators) domains.  Some studies of vulnerability assessments to flood include physical indicators (e.g., floodplain area, low elevation, proximity to river, etc.)  in their composite vulnerability index to reflect flood susceptibility or risk (Dewan, 2013; Islam et al., 2013; Roy and Blaschke, 2015; Hoque et al., 2019)

145 . In this study, however, the SHV index precludes those physical indicators, as flood hazard information (i.e., flood inundation) will be linked later through the impact assessment. Instead, we include a health domain uniquely reflecting flood-induced health risk that  has rarely been considered in previous studies (Ahsan and Warner, 2014; Gain et al., 2015; Rabby et al., 2019)

[revised manuscript text omitted]

**6 Conclusions**

In this study, we assess three domain vulnerabilities and a composite social-health vulnerability for all of Bangladesh. Results indicate that vulnerable zones exist in the northwest riverine areas, northeast floodplains, and southwest region, potentially affecting 42 million people (26% of total population). Then, for the first time, we incorporate predictive information  (flood forecast and satellite inundation) into vulnerability and validate it with  the recent catastrophic August 2017 flood event. Our findings suggest that the both forecast and satellite-based vulnerabilities can better inform observed flood impacts.

Compared to conventional and existing approaches, the approach and vulnerability measures developed here have unique advantages and contributions. First, our approach covers all of Bangladesh at a high resolution (Upazila). While local studies provide more specific analyses (e.g., key vulnerability factors within a city), our scale and resolution can support national-level assessment and management when massive monsoon floods affect most of the country. Second, given the lead-time of flood forecasts, the predicted vulnerability can dynamically anticipate flood impacts and actively support pre- and post-disaster management rather than being  applied as static vulnerability data as conventional approaches provide. We also demonstrate, through a validation, that the thematic (domain) vulnerability can better estimate a particular aspect of flood impacts. This can potentially facilitate tailored management actions, such as prioritizing different resources (e.g., foods, cash, medical supplies, volunteers, etc.), for the given location.

In order to enhance the quality of this approach, the validation process can be updated and improved with additional data, indicators, and flood records across the country to enhance management practices. Especially, more accurate post-disaster impact records with diverse variables at the local level (e.g., Upazila scale) may improve future vulnerability and risk assessments and impacts prediction. Flood forecasts have clear value, however producing local scale information may pose challenges in countries with limited resources; existing global scale forecasts may be able to fill this role and should be evaluated (Alfieri et al., 2013; Emerton et al., 2018). Understanding the prospects for extending forecast lead-times is also warranted, and may facilitate more proactive disaster management practices (Coughlan de Perez et al., 2015). Finally, integrating more physical flood information and models to estimate the affected population may enhance flood impact predictions.

385     Although this study includes Bangladesh as a single case study country, the proposed approach may be transferable to different countries where sufficient data are available. Specifically, given that this approach focuses on social and health vulnerability, the demographic and socio-economic components of vulnerability require at least sub-national level census data. In addition, data on observed flood impacts (i.e., post-disaster reports) are required to validate this approach at other locations. Integrating this approach systematically with a flood forecast system, such as a web-based online tool,

390 may be of further value to international and local disaster managers. Overall, this study provides groundwork for the development of a multi-sectoral (flood and health) risk warning system. Actionable flood and health risk predictions can radically improve existing disaster management practices of NGOs and other private and public organizations and save lives and resources by providing advanced preparedness and

395 response strategies.

*Data availability.* The original data of vulnerability indicators are publicly available (see Section 2.1). The processed vulnerability indicators can be requested from the corresponding author.

*Author contributions.* The paper and its methodology were conceptualized and developed by DL and PB; the data processing, analyses, and visualization were carried out by DL. The original draft was prepared by DL and PB; further reviewing and editing was carried out by DL,

400 PB, JP, and HA.

*Competing interests.* The authors declare that they have no conflict of interest.

*Acknowledgements.* The lead author would like to thank Mr. Arifuzzaman Bhuyan at FFWC for providing flood forecast data.

*Financial support.* This research is partially funded by the Global Health Institute (GHI) of the University of Wisconsin-Madison, the Wisconsin Alumni Research Foundation (WARF) – UW2020, and NASA project NNX17AC50G.

[revised manuscript text omitted]

---

## Referee Comment (RC2) · Anonymous Referee #2 · 20 Apr 2021

I found this manuscript well written and structured, and this easily allow the understanding of the complex multidata analyses presented. The focus is on an interesting topic and the analysis of people health vulnerability is new in the framework of countries, as Bangladesh, where floods impact on very large amount of people. I have no major change to suggest because the manuscript can be easily understood from the beginning to the end. I suggest the following minor modifications: 1) The authors could include in the introduction some references to the analysis of flood mortality in other countries, and to highlight the difference that the indicators used can assume according to the socio-economic framework of the flooded area. 2) It must be stressed that the "validation" is based on a single case of flood, in order to offer to the reader the clear perception of the weight of the results. 3) Maps need some geographical grid, otherwise thy are not easy to understand for people who do not know the regions. 4) Line 100: maybe "coarse" instead of course? 5) Figure 2: I suggest to modify the names of indicators, to be more clear (i.e.: Pfemale or P-FEMALE or something similar)

---

## Author Response (AR1)

**Reply to RC1: 'Manuscript revision', Anonymous Referee #1, 27 Dec 2020**

Authors' replies are in blue color and revised sentences are in italics.

Generally, the manuscript is very well written and provides an innovative and simple approach to assess vulnerability. The structure of the paper is clear. Data and methods are well explained, and possible lacks are explained in detail. The title is appropriate and reflects what is reported in the document. I have few minor revisions to propose. In my opinion, the authors should improve and clarify some terms used in the manuscript such as "indicator", "index", and "score". I suggested specifying the meaning of these terms referring to the existing bibliography.

1) We thank the anonymous reviewer for the positive comments and further critical comments that we believe will enhance the overall quality of the manuscript. As you suggested, we now better clarify the terms "indicator", "index", and "score". Specifically, we have provided the following sentence at Line 68 (at the beginning of Data section):

   *In this study, as is generally adopted in the literature, an indicator represents an individual variable, an index represents a composite vulnerability, and a score indicates the value of an indicator or index (Birkmann, 2006; Fekete, 2009).*

The figures are for the most part clear and well organized but cartographic elements such as the North and scale bar should be added at least for the first one, as well as the presence of UTM coordinates on grids.

2) We agree with the reviewer. We have added a location map showing flood forecast and inundation with key regions explained in the manuscript (e.g., Dhaka, Chittagong, and Haor basin). Please find the Figure 1 below.

The references are relevant and sufficient. However, some statements need to be supported by references, in those cases indicated in the revision, they must be added.

3) We have added additional references. Please find our specific answers to your comments below.

Here below specific points and minor issues.

Abstract
The abstract is clear and short. I do not have any suggestions

Introduction
For this chapter, I have just one suggestion. The authors produce a good explanation of the work and the importance of it in a Bangladesh contest. Moreover, sentences are followed by proper references. My suggestion regards the absence of a location map. For the readers, it would be very useful to have a location map reporting the altitude and some locations

mentioned in the manuscript ("Dhaka", "Chittagong", "Haor basin"). This can guide the reader to focus on the more interesting and significant areas of the work itself.

4) Thanks for your comment. We have provided a location map (Figure 1).

Line 60: "social-health vulnerability (SHV) indicator". Term "indicator is correct? In figure 4 the authors use SHV "index". Please clarify the question of the terms as explained in the general comments.

5) We have corrected this to read: "social-health vulnerability (SHV) index".

Data
The data sources are well explained. I do not have any suggestions.

Methods
Line 139: Please add Index or indicator to "SHV"

6) We have changed "SHV" to "SHV index" throughout the manuscript.

Line 140 to 142. In my opinion, this sentence has to be reformulated and clarified with more detail.

7) To clarify, we have changed Lines 140-142 to:

*Some studies of vulnerability assessments to flood include physical indicators (e.g., floodplain area, low elevation, proximity to river, etc.) in their composite vulnerability index to reflect flood susceptibility or risk (Dewan, 2013; Islam et al., 2013; Roy and Blaschke, 2015, Hoque et al., 2019). In this study, however, the SHV index precludes those physical indicators, as flood hazard information (i.e., flood inundation) will be linked later through the impact assessment.*

Line 144: "Previous studies". In my opinion relevant references are needed

8) Thanks for this comment. We have added references to Lines 142-144:

*Instead, we include a health domain uniquely reflecting flood-induced health risk that has rarely been considered in previous studies (Ahsan and Warner 2014, Gain et al., 2015, Rabby et al., 2019) (Table 1).*

Line 148: A citation is needed after the word "hazard"

9) We have added references to Line 148:

*Comparatively, the coping capacity domain represents the ability to cope with or adapt to the hazard (Birkmann, 2006; Villagrán de Léon, 2006).*

Line 150: Some citations are needed after the words "etc"

10) We have added references to Line 150:

*In the literature, coping capacity indicators are surveyed for the local region, or proxy data from the census are used, such as households with communication devices and vehicles, literacy rates, education levels, etc (Dewan 2015; Roy and Blaschke, 2015).*

Results
The presentation of the results is satisfactory, although in some sentences are too simplified. For example, at section 4.3 the sentence between line 282 and 285 should be better explained and developed. The overestimation due to insufficient data quality should be verified more in detail, overall if the authors declare that the proposed approach is "transferable and easily adapted to different countries". The data quality is not analysed with enough detail. I suggest to spend some lines to explain the possible consequences of the data quality or missing data on the proposed approach.

11) Thanks for pointing this out. To clarify, Lines 282-287 have been changed to:

*This overestimation of the affected population is likely attributable to the simplified approaches and a lack of data on flood management and properties. For example, the two approaches adopted here do not consider the level of flood protection (e.g., embankment, levee, early warning, etc.) but rather assume that all regions have an equivalent level of protection and management. Furthermore, the current flood forecasts and satellite inundation information do not provide specific physical flood properties, such as duration of the flood, which is a key factor in increasing flood impacts, as indicated in the post-flood reports.*

12) Also, Line 365 has been changed to:

*Although this study includes Bangladesh as a single case study country, the proposed approach may be transferable to different countries where sufficient data are available. Specifically, given that this approach focuses on social and health vulnerability, the demographic and socio-economic components of vulnerability require at least sub-national level census data. In addition, data on observed flood impacts (i.e., post-disaster reports) are required to validate this approach at other locations.*

Discussion and conclusion
In my opinion this chapter needs to be rearranged. Discussion and conclusion should be divided in two different chapters. Furthermore, in the discussion chapter, the authors should compare their own approach with other international published approaches. This suggestion is made in order to highlight the quality of the approach shown and which are the advantages compared with other existing approaches.

13) Thanks for your comment. We note that no international published approaches have assessed vulnerability for all of Bangladesh at this resolution. Instead, we have highlighted the new contributions of our approach in comparison with conventional approaches. To clarify, Lines 351-356 have been changed to:

*Compared to conventional and existing approaches, the approach and vulnerability measures developed here have unique advantages and contributions. First, our approach covers all of Bangladesh at a high resolution (Upazila). While local studies provide more specific analyses (e.g., key vulnerability factors within a city), our scale and resolution can support national-level assessment and management when massive monsoon floods affect most of the country. Second, given the lead-time of flood forecasts, the predicted vulnerability can dynamically anticipate flood impacts and actively support pre- and post-disaster management rather than being applied as static vulnerability data as conventional approaches provide. We also demonstrate, through a validation, that the thematic (domain) vulnerability can better estimate a particular aspect of flood impacts. This can potentially facilitate tailored management actions, such as prioritizing different resources (e.g., foods, cash, medical supplies, volunteers, etc.), for the given location.*

14) Also, we have separated and added a Conclusion section as follows:

*6 Conclusions*

*In this study, we assess three domain vulnerabilities and a composite social-health vulnerability for all of Bangladesh.  Results indicate that vulnerable zones exist in the northwest riverine areas, northeast floodplains, and southwest region, potentially affecting 42 million people (26% of total population). Then, for the first time, we incorporate predictive information (flood forecast and satellite inundation) into vulnerability and validate it with the recent catastrophic August 2017 flood event. Our findings suggest that the both forecast and satellite-based vulnerabilities can better inform observed flood impacts.*

*Compared to conventional and existing approaches, the approach and vulnerability measures developed here have unique advantages and contributions. First, our approach covers all of Bangladesh at a high resolution (Upazila). While local studies provide more specific analyses (e.g., key vulnerability factors within a city), our scale and resolution can support national-level assessment and management when massive monsoon floods affect most of the country. Second, given the lead-time of flood forecasts, the predicted vulnerability can dynamically anticipate flood impacts and actively support pre- and post-disaster management rather than being applied as static vulnerability data as conventional approaches provide. We also demonstrate, through a validation, that the thematic (domain) vulnerability can better estimate a particular aspect of flood impacts. This can potentially facilitate tailored management actions, such as*

*prioritizing different resources (e.g., foods, cash, medical supplies, volunteers, etc.), for the given location.*

*In order to enhance the quality of this approach, the validation process can be updated and improved with additional data, indicators, and flood records across the country to enhance management practices. Especially, more accurate post-disaster impact records with diverse variables at the local level (e.g., Upazila scale) may improve future vulnerability and risk assessments and impacts prediction. Flood forecasts have clear value, however producing local scale information may pose challenges in countries with limited resources; existing global scale forecasts may be able to fill this role and should be evaluated (Alfieri et al., 2013; Emerton et al., 2018). Understanding the prospects for extending forecast lead-times is also warranted, and may facilitate more proactive disaster management practices (Coughlan de Perez et al., 2015). Finally, integrating more physical flood information and models to estimate the affected population may enhance flood impact predictions.*

*Although this study includes Bangladesh as a single case study country, the proposed approach may be transferable to different countries where sufficient data are available. Specifically, given that this approach focuses on social and health vulnerability, the demographic and socio-economic components of vulnerability require at least sub-national level census data. In addition, data on observed flood impacts (i.e., post-disaster reports) are required to validate this approach at other locations. Integrating this approach systematically with a flood forecast system, such as a web-based online tool, may be of further value to international and local disaster managers. Overall, this study provides groundwork for the development of a multi-sectoral (flood and health) risk warning system. Actionable flood and health risk predictions can radically improve existing disaster management practices of NGOs and other private and public organizations and save lives and resources by providing advanced preparedness and response strategies.*

Moreover the sentence "The proposed approach is transferable and easily adapted to different countries to assess vulnerabilities ", is very strong. This approach is used just for a single case of study and in a specific country. I think that this sentence should be reformulated due to the fact that the method has different constrains and should be clear in what conditions of data and knowledge it could applied.

15) We have changed this sentence. Please refer to our reply #12.

References

Ahsan, M. N. and Warner, J.: The socioeconomic vulnerability index: A pragmatic approach for assessing climate change led risks-A case study in the south-western coastal Bangladesh, International Journal of Disaster Risk Reduction, 8, 32–49, https://doi.org/10.1016/j.ijdrr.2013.12.009, 2014.

Alfieri, L., Burek, P., Dutra, E., Krzeminski, B., Muraro, D., Thielen, J., and Pappenberger, F.: GloFAS – global ensemble streamflow forecasting and flood early warning, Hydrology and Earth System Sciences, 17, 1161–1175, https://doi.org/10.5194/hess-17-1161-2013, 2013.

Birkmann, J.: Measuring vulnerability to natural hazards: Towards disaster resilient societies, vol. 02, United Nations University, New York, 2006.

Coughlan de Perez, E., van den Hurk, B., van Aalst, M. K., Jongman, B., Klose, T., and Suarez, P.: Forecast-based financing: an approach for catalyzing humanitarian action based on extreme weather and climate forecasts, Natural Hazards and Earth System Science, 15, 895–904, https://doi.org/10.5194/nhess-15-895-2015, 2015.

Dewan, A. M.: Floods in a megacity: Geospatial techniques in assessing hazards, risk and vulnerability, in: Floods in a Megacity: Geospatial Techniques in Assessing Hazards, Risk and Vulnerability, chap. 6, pp. 1–199, Springer Netherlands, Dordrecht, https://doi.org/10.1007/978-94-007-5875-9, 2013.

Emerton, R., Zsoter, E., Arnal, L., Cloke, H. L., Muraro, D., Prudhomme, C., Stephens, E. M., Salamon, P., and Pappenberger, F.: Developing a global operational seasonal hydro-meteorological forecasting system: GloFAS-Seasonal v1.0, Geoscientific Model Development, 11, 3327–3346, https://doi.org/10.5194/gmd-11-3327-2018, 2018.

Fekete, A.: Validation of a social vulnerability index in context to river-floods in Germany, Natural Hazards and Earth System Science, 9, 393–403, https://doi.org/10.5194/nhess-9-393-2009, 2009.

Gain, A. K., Mojtahed, V., Biscaro, C., Balbi, S., and Giupponi, C.: An integrated approach of flood risk assessment in the eastern part of Dhaka City, Natural Hazards, 79, 1499–1530, https://doi.org/10.1007/s11069-015-1911-7, 2015.

Hoque, M. A. A., Tasfia, S., Ahmed, N., and Pradhan, B.: Assessing spatial flood vulnerability at kalapara upazila in Bangladesh using an analytic hierarchy process, Sensors, 19, 1302, https://doi.org/10.3390/s19061302, 2019.

Islam, A. N., Deb, U. K., Al Amin, M., Jahan, N., Ahmed, I., Tabassum, S., Ahamad, M. G., Nabi, A., Singh, N. P., Kattarkandi, B., and Bantilan, C.: Vulnerability to Climate Change: Adaptation Strategies and Layers of Resilience Quantifying Vulnerability to Climate Change in Bangladesh, Tech. rep., International Crops Research Institute for the Semi-Arid Tropics (ICRISAT), Telangana, India, http://oar.icrisat.org/8117/, 2013.

Rabby, Y. W., Hossain, M. B., and Hasan, M. U.: Social vulnerability in the coastal region of Bangladesh: An investigation of social vulnerability index and scalar change effects, International Journal of Disaster Risk Reduction, 41, 101 329, https://doi.org/10.1016/j.ijdrr.2019.101329, 2019.

Roy, D. C. and Blaschke, T.: Spatial vulnerability assessment of floods in the coastal regions of Bangladesh, Geomatics, Natural Hazards and Risk, 6, 21–44, https://doi.org/10.1080/19475705.2013.816785, 2015.

Villagrán de Léon, J. C.: Vulnerability: a conceptional and methodological review, UNU Institute for Environment and Human Security (UNU-EHS), https://collections.unu.edu/view/unu:1871, 2006.

[Figure]

Figure 1. (left) FFWC's flood forecast issued on Aug-16, 2017 and (right) Sentinel-1 based satellite flood inundation for the August 2017 flood event. The borderline represents the boundary of divisions.

***Reply to RC2: 'manuscript revision', Anonymous Referee #2, 20 Apr 2021***

Authors' replies are in blue color and revised sentences are in italics.

I found this manuscript well written and structured, and this easily allow the understanding of the complex multidata analyses presented. The focus is on an interesting topic and the analysis of people health vulnerability is new in the framework of countries, as Bangladesh, where floods impact on very large amount of people. I have no major change to suggest because the manuscript can be easily understood from the beginning to the end. I suggest the following minor modifications:

> We thank the anonymous reviewer for the positive comments and further critical comments that we believe have enhanced the overall quality of the manuscript.

1) The authors could include in the introduction some references to the analysis of flood mortality in other countries, and to highlight the difference that the indicators used can assume according to the socio-economic framework of the flooded area.

> Thanks for your comment. We added the following paragraph as the first paragraph of the introduction.

> *Flood-induced mortality, one of the most telling statistics of flood impacts, has been studied extensively in conjunction with environmental and socio-economic factors. For example, Kundzewicz and Takeuchi (1999) demonstrate the relationship between economic losses per death and overall national wealth for the most severe flood events of the 1990s. Kundzewicz and Kundzewicz (2005) also emphasize that flood-related mortality is indirectly related to wealth level and instead is more directly related to social and health factors and perceptions of flood risk, based on information from flood victims in Poland in 1997. According to Jonkman and Vrijling (2008), the primary causes of flood-related mortality are a lack of warning, inability to reach shelter, building collapse, flood level and velocity, and impacts on children and elderly. Doocy at al. (2013) review global flood fatality data during 1980-2009 and related articles, concluding that socio-demographic factors such as population growth, urbanization, land use change, disaster warning systems, and response capacity all contribute to flood mortality.*

2) It must be stressed that the "validation" is based on a single case of flood, in order to offer to the reader the clear perception of the weight of the results.

> We agree with the reviewer. To emphasize the need for further validation of the proposed approach, Lines 356-359 have been changed to:

> *We note that the proposed framework has been validated with a single observed flood event, and additional validation using more flood events is warranted. Furthermore, the validation process could be improved using up-to-date data, indicators, and flood records across the country to enhance management practices. Specifically, more*

*detailed post-disaster impact records at the local level (e.g., Upazila scale) may improve future vulnerability and risk assessments and impacts prediction.*

3) Maps need some geographical grid, otherwise they are not easy to understand for people who do not know the regions.

We have added a location map indicating flood forecast and inundation areas with geographical grids. We have also better presented key regions in the manuscript (e.g., Dhaka, Chittagong, and Haor basin). Please see Figure 1 below. We have opted to not modify all maps, as we believe Figure 1 will help readers become familiar with Bangladesh's geographical position and extents.

4) Line 100: maybe "coarse" instead of course?

Thanks for correcting this. We have changed it to "coarse."

5) Figure 2: I suggest to modify the names of indicators, to be more clear (i.e.: Pfemale or P-FEMALE or something similar)

Thanks for your comment. We have changed the names of the indicators to make them more intuitive. Please see Table 1 below. Also, the names of the indicators in the manuscript and Figure 3 have been properly changed as well.

Table 1. Changes in the names of the indicators

| Before | After |
|---|---|
| PAGEWEAK | P_WEAK-AGE |
| PFEMALE | P_FEMALE |
| PDISABL | P_DISABLE |
| PRURAL | P_RURAL |
| PWEAKBUILT | P_WEAK-HOUSE |
| PNOWATER | P_WATER-SUPPLY |
| PNOSANITARY | P_SANITATION |
| PNOELEC | P_ELECTRICITY |
| PLITERACY | P_LITERACY |
| PETHNIC | P_ETHNIC |
| PRENT | P_RENT |
| PNOPRIEDU | P_EDUCATION |
| PPOOR | P_POOR |
| PAGRICULT | P_AGRICULTURE |
| PNOEMPLOY | P_EMPLOYMENT |
| PDISEASE | P_DISEASE |
| PDIARRHEA | P_DIARRHEA |
| PDISEASEDWATER | P_WATER-DISEASE |
| NHOSPITALBED | N_HOSPITAL-BED |
| NPHYSICIAN | N_PHYSICIAN |
| PAFFTHOUS | P_HOUSE-AFFECTED |
| PNOSCHOOL | P_CHILD-SCHOOL |
| PNOPREPARED | P_PREPAREDNESS |
| PPERCEPTION | P_PERCEPTION |
| PSUPPORT | P_SUPPORT |
| DAMAGERATIO | R_DAMAGE-INCOME |

**References**

Kundzewicz, Z. and Kundzewicz, W.: Mortality in flood disasters, in: Extreme weather events and public health responses, pp. 197–206, Springer, 2005.

Kundzewicz, Z. W. and Takeuchi, K.: Flood protection and management: quo vadimus?, Hydrological Sciences Journal, 44, 417–432, 1999.

Jonkman, S. and Vrijling, J.: Loss of life due to floods, Journal of Flood Risk Management, 1, 43–56, 2008.

Doocy, S., Daniels, A., Murray, S., and Kirsch, T. D.: The human impact of floods: a historical review of events 1980-2009 and systematic literature review, PLoS currents, 5, 2013.

[Figure]

Figure 1. (left) FFWC's flood forecast issued on Aug-16, 2017 and (right) Sentinel-1 based satellite flood inundation for the August 2017 flood event. The borderline represents the boundary of divisions.

[Figure]

Figure 3.Cross-correlation matrix of the selected indicators calculated at Upazila-level unless followed by an asterisk (district-level.) Theplus sign indicates a statistically significant correlation (p <0.05).